# Surgery for Ulcerative Colitis in the White British and South Asian Populations in Selected Trusts in England 2001–2020: An Absence of Disparate Care and a Need for Specialist Centres

**DOI:** 10.3390/jcm11174967

**Published:** 2022-08-24

**Authors:** Affifa Farrukh, John Francis Mayberry

**Affiliations:** Nuffield Hospital, Leicester LE5 1HY, UK

**Keywords:** ulcerative colitis, surgery, ileostomy, ileo-anal anastomosis, ethnicity

## Abstract

Over the last decade, there has been extensive evidence that patients with inflammatory bowel disease from minority communities in the UK receive less than optimal care. In none of the studies has the role of surgery in the management of acute and severe ulcerative colitis been considered in any detail. A freedom of information (FOI) request was sent to 14 NHS Trusts in England, which serve significant South Asian populations. Details of the type of surgery patients from the South Asian and White British communities received between 2021 and 2020 were requested. Detailed responses were obtained from eight Trusts. Four hundred and ten White British patients underwent surgery for ulcerative colitis over this period at these eight Trusts, together with 67 South Asian patients. There was no statistically significant difference in the distribution across the types of surgery undergone by the two communities overall (χ^2^ = 1.3, ns) and the proportions who underwent an ileo-anal anastomosis with pouch (z = −1.2, ns). However, within individual trusts, at the University Hospital Southampton NHS Foundation Trust, a significantly greater proportion of South Asian patients had an ileo-anal anastomosis with pouch compared to White British patients. At Cambridge University Hospitals NHS Foundation Trust, all 72 patients who underwent surgery for ulcerative colitis were White British. This study has shown that, in general, for patients with a severe flare of ulcerative colitis where medical treatment has failed and surgery is warranted, the nature of the procedures offered is the same in the White British and South Asian communities. However, of concern is the number of units with low volume procedures. For most Trusts reported in this study, the overall number of Ileo-anal pouch anastomosis or anastomosis of ileum to anus procedures performed over a number of years was substantially below that required for a single surgeon to achieve competence. These findings reinforce the argument that inflammatory bowel disease surgery should be performed in a limited number of high-volume centres rather than across a wide range of hospitals so as to ensure procedures are carried out by surgeons with sufficient and on-going experience.

## 1. Introduction

Over the last decade, there has been extensive evidence that patients with inflammatory bowel disease from minority communities in the UK receive less than optimal care [1,2,3,4,5]. This has taken the form of less frequent consultations with senior clinicians, fewer investigations, more frequent discharge from follow-up clinics [3] and more limited access to expensive biologic therapies [1,2,4]. Such disparate care is not seen in all Trusts but is widespread and has been shown to effect patients who are from South Asian, Afro-Caribbean and Eastern European communities. In none of these studies has the role of surgery in the management of acute and severe ulcerative colitis been considered.

## 2. Method

A freedom of information (FOI) request was sent to the following 15 NHS Trusts in England which serve significant South Asian populations:Barking, Havering and Redbridge University Hospitals NHS TrustBarts Health NHS TrustBradford Teaching Hospitals NHS Foundation TrustCambridge University Hospitals NHS Foundation TrustEast Lancashire Hospitals NHS TrustNorth West Anglia Foundation TrustNorthern Care Alliance NHS Foundation TrustSandwell and West Birmingham NHS TrustThe Hillingdon Hospitals NHS Foundation TrustThe Royal Wolverhampton NHS TrustUniversity Hospitals Coventry and Warwickshire NHS TrustUniversity Hospitals of Derby and Burton NHS Foundation TrustUniversity Hospitals of Leicester NHS TrustUniversity Hospital Southampton NHS Foundation TrustWalsall Healthcare NHS Trust

The FOI request asked each hospital to provide information on the following types of surgery in the Trust between 1 January 2001 and 31 December 2020

Ileo-anal pouch anastomosis or anastomosis of ileum to anus (H042 and H043) (IPAA)Total colectomy and anastomosis of ileum to rectum (H051)Panproctocolectomy and ileostomy (HO41)

H041, H042, H043 and H051 are disease and procedure codes from ICD-10, which is the 10th revision of the International Statistical Classification of Diseases and Related Health Problems [6]. Data were not collected on the total number of patients with ulcerative colitis treated at each trust. Such data would have included outpatients who received medical treatment alone and these are not routinely collected by Trusts. Rather the purpose of the study was to consider whether patients from different ethnic groups who underwent surgery received similar operations or not. In addition, data on 30-day mortality rates, re-operation rates, leak and other complication rates were not requested. Such searches require more lengthy collection periods and often require access to data collected on local registers held by hospitals, surgical departments or individual surgeons. Within the Freedom of Information Act, such requests can be declined on the basis of cost and issues with lack of anonymity.

Trusts were asked to provide data for each quinquennium between 2001 and 2020 for two ethnic groups:White British (National code A)Asian (Pakistani, Indian, Bangladeshi origin) (National codes H, J and K)

A, H, J and K are ethnicity codes used by the NHS and defined in *The NHS Data Model and Dictionary* [7].

Data for each trust were summated for all the periods for which information was provided. A χ^2^ test was performed to compare the forms of treatment received by the two groups of patients using the Social Science Statistics calculator [8]. In addition, a z test was used to compare the rates for ileo-anal anastomosis in the two populations for individual trusts and overall.

## 3. Results

Detailed responses were obtained from eight Trusts. However, three of these Trusts were only able to provide data for part of the period 2001–2020 (See Table 1). Seven of the Trusts, Barking, Havering and Redbridge (FOI 7981), Barts NHS Trust (FOI-0225-22), Coventry and Warwick (FOI 1732), Northern Care Alliance (FOI 11854), Wolverhampton (FOI 100422) Hillingdon Hospitals (FOI 6782) and North West Anglia (FOI/2022/400), were unable to provide detailed breakdowns of their practice due to small numbers and concerns about patient anonymity.

Four hundred and ten White British patients underwent surgery for ulcerative colitis over this period at the 8 Trusts, who provided a detailed response, together with 67 South Asian patients (Table 1). There was no statistically significant difference in the distribution across the types of surgery undergone by the two communities overall (χ^2^ = 1.3, ns) and the proportions who underwent an ileo-anal anastomosis with pouch (z = −1.2, ns). Although the value of statistical analysis within individual Trusts was limited by small numbers an analysis was performed to identify any Trusts which appeared to lie outside of the overall practice. Within individual trusts, at the University Hospital Southampton NHS Foundation Trust, a significantly greater proportion of South Asian patients had an ileo-anal anastomosis with pouch compared to White British patients (See Table 1). Indeed, all seven South Asian patients had a pouch procedure.

In order to assess the validity of the data obtained by Freedom of Information requests, the response from University Hospitals of Leicester NHS Trust was further scrutinised. The incidence of ulcerative colitis in the South Asian community is higher than in the White British community, and the disease is more aggressive in second generation migrants [9,10]. The population served by the Trust covers Leicester, Leicestershire and Rutland, with the South Asian community forming 17% of the population compared to 83% White British, excluding other communities [11]. If the disease is of comparable frequency and severity, then the expected number of patients in a group of 187, 32 would be South Asian. The group actually included 36 South Asian patients (z = −0–54, ns). Based on the data available from a study of surgical records of patients with ulcerative colitis in Leicester covering the period 1997 to 2007 [12], the expected number of South Asian patients would have been 37 (z = −0.13, ns). These latter data reflect differences in disease severity and yield an even closer match.

However, the results do raise various areas of concern. At Cambridge University Hospitals NHS Foundation Trust, all 72 patients who underwent surgery for ulcerative colitis were White British and none were South Asian. The very low overall number of patients reported by Barking, Havering and Redbridge, Coventry and Warwick, Northern Care Alliance, Wolverhampton, Hillingdon Hospitals and North West Anglia raises questions as to whether such surgery should occur at these centres or patients should be referred to a more experienced centre.

## 4. Discussion

This study has shown that, in general, for patients with a severe flare of ulcerative colitis where medical treatment has failed and surgery is warranted, the nature of the procedures offered is the same in the White British and South Asian communities. The choice includes a traditional ileostomy and proctectomy, with an associated significant impact on body image and the need to use stoma appliances. Both have been shown to have a significant impact for South Asian patients [13]. However, the surgery is relatively straightforward. Colectomy with an ileo-rectal anastomosis is, perhaps, the simplest procedure with a good continence outcome, but the patient remains at risk of colorectal cancer. It is for this reason that the procedure has never gained popularity in the UK, and in this study, only 10 of 477 patients underwent this procedure. An ileo-anal anastomosis is associated with an acceptable body image, but patients have a high frequency of defaecation and may develop pouchitis. It is a more complex procedure and ideally should be performed in centres with a large case load. The check on the data from an FOI in Leicester would indicate that the findings are robust with slightly more patients undergoing surgery, consistent with the greater frequency of the disease and its being more severe in South Asians. FOI data have previously been shown to be robust when considering data collected about the management of inflammatory bowel disease [2], and this study is consistent with such findings.

A recent study of Hospital Episode statistics has shown that Asian patients with ulcerative colitis are likely to experience a delay in colectomy surgery when admitted as an emergency [14]. In 1992, a study of Asian ostomists reported that only half of them had been able to discuss their management with a doctor or nurse who spoke the same language, whereas stoma care nurses thought that the service at that time was adequate [14]. In a later study of 107 patients with ulcerative colitis or familial adenomatous polyposis from Leicester, in 2009, complication rates and long-term outcome were comparable for White British and South Asian patients following a restorative proctocolectomy, with the exception of pouchitis, which was commoner in the latter group [12]. However, a separate study showed that language issues were clearly linked to a poorer quality of life and consistent with the views expressed by the ostomists [15]. Clearly, the fact that South Asian and White British patients experience the same types of surgery does not remove the need to explain their nature and consequences, such that these issues are understood by patients.

Of concern in this study is the number of units with low volume procedures. A study from the Cleveland Clinic, USA has shown that trainee staff undertaking stapled IPAA surgery only showed an improvement in the pouch failure rate following an initial training period of 23 cases versus 40 cases for senior staff [16]. The learning curve for hand-sewn IPAA surgery was quantified only for senior staff, who attained adequate results following an initial period of 31 procedures. This formed the basis for the recommendation in the 2019 British Society of Gastroenterology Guidelines that:“**Statement** **22.** We suggest that pouch surgery should be performed in specialist high-volume referral centres (GRADE: weak recommendation, low-quality evidence. Agreement 97.4%) [17].”

In most Trusts reported in this study, the overall number of IPAA procedures performed over a number of years was substantially below that required for a single surgeon to achieve competence. These findings reinforce the argument that inflammatory bowel disease surgery should be performed in a limited number of high-volume centres rather than across a wide range of hospitals in order to ensure that procedures are carried out by surgeons with sufficient and on-going experience.

Reports from two trusts are worthy of comment. In Southampton, all 7 patients underwent an IPAA procedure, and in Cambridge, no South Asian patients underwent surgery for acute colitis compared to 72 White British patients over a 20-year period. FOI searches do not allow an explanation for these outliers, but such findings should prompt consideration by the relevant Trusts as to their approach to minority community patients.

In conclusion, although severely ill patients with ulcerative colitis receive the same range of surgical options regardless of whether they are of South Asian or White British origin there are trusts whose practice is outlying and the reasons are not known. In addition, this study has demonstrated the limited experience of such procedures in some trusts, all trusts were selected because they served significant South Asian communities and such a finding may not be more general. However, there is serious need to consider concentrating surgical expertise in high-volume centres.

## Figures and Tables

**Table 1 jcm-11-04967-t001:** Types of Surgery for Ulcerative Colitis in the White British and South Asian Populations in Scheme 2001–2020.

Trust	Period	Ileoanal Anastomosis	Ileorectal Anastomosis	Ileostomy	Number	Significance
**Southampton**	2001–2020					
White		48	0	36	84	
Asian		7	0	0	7	z = −2.23, *p* < 0.03
**Leicester**	2001–2020					
White		59	6	86	151	
Asian		13	0	23	36	ns
**Sandwell & West Birmingham**	2001–2020					
White		3	0	8	11	
Asian		0	0	1	1	ns
**Walsall**	2014–2015					
White		1	0	1	2	
Asian		0	0	0	0	ns
**Derby & Burton**	2001–2020					
White		4	1	17	22	
Asian		1	0	1	2	ns
**East Lancashire Hospitals**	2006–2020					
White		7	0	16	23	
Asian		0	0	0	0	Ns
**Bradford**	2007–2020					
White		26	1	18	45	
Asian		13	0	8	21	Ns
**Cambridge**	2001–2020					
White		28	2	42	72	
Asian		0	0	0	0	
**Total**						
White		176	10	224	410	
Asian		34	0	33	67	ns

## Data Availability

These links are listed in the FOI Request References.

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
