# Peer review of "Surgery for Ulcerative Colitis in the White British and South Asian Populations in Selected Trusts in England 2001–2020: An Absence of Disparate Care and a Need for Specialist Centres"

_jcm, 2022, doi:10.3390/jcm11174967_

Round 1

Reviewer 1 Report

Minor points:

1.  Authors need to describe how many patients with UC were registered in each NHS Trust as the basis of statistical analysis. (Table 1)

2. IPAA, IRA, and Ileostomy for patients with UC should be shortly explained about the clinical merit and demerit in the Method or Discussion.

3.  Approval of this study by the ethics committee for clinical research is recommended.

Author Response

  1. Authors need to describe how many patients with UC were registered in each NHS Trust as the basis of statistical analysis. (Table 1)A comment has been added to clarify that the study is dealing with patients undergoing surgery for ulcerative colitis. No Trust holds data on the overall number of patients with ulcerative colitis in their care. Data are not routinely collected on outpatients with IBD and if it is then it is part of a research protocol and such data are excluded from disclosure by the Freedom of Information Act

2. IPAA, IRA, and Ileostomy for patients with UC should be shortly explained about the clinical merit and demerit in the Method or Discussion.

This has  been added in the Discussion

3.  Approval of this study by the ethics committee for clinical research is recommended.

Ethical approval is not required for data obtained by Freedom of Information searches. Such data are freely available to anyone and can be requested by anyone and  this is enshrined in the Act. Once such a search is conducted the data is immediately available to anyone else who makes an inquiry. Hence we have provided the index numbers for the searches. Many papers on  FOI  searches have been published over the years and in none has approval been sought from an ethics committee. The nature of the data available is regulated by the Act and this is overseen by the Information Commissioner

Reviewer 2 Report

It is well recognized that high volume centers for specialized surgery will outperform those with low volume, low expertise. This has been published across most if not all specialties. The review article focuses on this concept. However, it does shed new light that there is no obvious ethnic or racial bias between the study populations when allocating surgical resources to the management of ulcerative colitis within the examined Trusts.

The authors question whether specialized surgery should occur at the smaller hospital locations due to their overall low volume. It would be of interest to have a point of reference to which these smaller hospitals are being compared to, especially for readers outside of the UK. For example, including nationally accepted outcome standards in terms of 30 day mortality, leak rate, etc and comparing these to the hospitals in this data set may help paint a better picture as to how far, or close, they perform to the standard of care as this may support their claim that specialized surgery should not be performed in some of the locations in the study. However, I am not sure whether this information was available from the FOI request to these sites of interest; if the information is available, it would be a great addition to the manuscript.

I do question whether there are enough N to run a statistical analysis in some of the data points used, such as Walsall, Derby & Burton, East Lancashire hospitals. potentially just reporting the raw numbers is more apporpriate. 

Authors may consider revisiting the article title. I believe the manuscript has two main points – there is no racial disparity within the allocation of surgical resources, and that complex surgery should be reserved for high volume centers; and neither of these main take home messages are incorporated into the title.

Other minor edits/suggestions:

Table. Is the FOI request number in the main study table relevant to the readers? The formatting is not uniform (two vs three decimal points, one entry has “/” in lieu of a “.”. Unclear if these are typos or accurate FOI numbers. Additionally, the table could be easier to follow if the total patient N for every Trust was shown for both white and Asian populations, rather than a gross total at the end of the column (it helps to see how 

Author Response

  1. Including nationally accepted outcome standards in terms of 30 day mortality, leak rate, etc and comparing these to the hospitals in this data set may help paint a better picture as to how far, or close, they perform to the standard of care as this may support their claim that specialized surgery should not be performed in some of the locations in the study. However, I am not sure whether this information was available from the FOI request to these sites of interest; if the information is available, it would be a great addition to the manuscript.

         Unfortunately this information was not available from the FOI. It was not  requested as FOI searches can be refused on the grounds that the time spent collecting the data would be too great and too costly and that some of these data would be research based and the FOI Act gives this as an exclusion cause. A comment has been entered into the text to confirm this.

2. I do question whether there are enough N to run a statistical analysis in some of the data points used, such as Walsall, Derby & Burton, East Lancashire hospitals. potentially just reporting the raw numbers is more appropriate. 

We accept this is a valid criticism. We have now entered the raw numbers in the Table and explained why despite the small numbers we were looking for centres lying outside the normal range

3. Authors may consider revisiting the article title. I believe the manuscript has two main points – there is no racial disparity within the allocation of surgical resources, and that complex surgery should be reserved for high volume centers; and neither of these main take home messages are incorporated into the title.

We have changed the title

4. Table. Is the FOI request number in the main study table relevant to the readers? The formatting is not uniform (two vs three decimal points, one entry has “/” in lieu of a “.”. Unclear if these are typos or accurate FOI numbers.

We have removed the FOI request numbers to the end of the paper. Their value is that they are raw data and anyone can request it from the Trust and providing the number will assist in the search. There is no uniform numbering system - hence the different structure of the references

4. Additionally, the table could be easier to follow if the total patient N for every Trust was shown for both white and Asian populations, rather than a gross total at the end of the column 

We have placed such a column in the Table